# Addressing Negative Transfer in Diffusion Models

**Hyojun Go**[1*]   **JinYoung Kim**[1*]   **Yunsung Lee**[2*]   **Seunghyun Lee**[3*]
**Shinhyeok Oh**[3]   **Hyeongdon Moon**[4]   **Seungtaek Choi**[5†]
Twelvelabs[1]   Wrtn Technologies[2]   Riiid[3]   EPFL[4]   Yanolja[5]
{gohyojun15, seago0828}@gmail.com[1], sung@wrtn.io[2] , {seunghyun.lee
shinhyeok.oh}@riiid.co[3], hyeongdon.moon@epfl.ch[4], seungtaek.choi@yanolja.com[5]

## Abstract

Diffusion-based generative models have achieved remarkable success in various domains. It trains a shared model on denoising tasks that encompass different noise levels simultaneously, representing a form of multi-task learning (MTL). However, analyzing and improving diffusion models from an MTL perspective remains under-explored. In particular, MTL can sometimes lead to the well-known phenomenon of *negative transfer*, which results in the performance degradation of certain tasks due to conflicts between tasks. In this paper, we first aim to analyze diffusion training from an MTL standpoint, presenting two key observations: **(O1)** the task affinity between denoising tasks diminishes as the gap between noise levels widens, and **(O2)** negative transfer can arise even in diffusion training. Building upon these observations, we aim to enhance diffusion training by mitigating negative transfer. To achieve this, we propose leveraging existing MTL methods, but the presence of a huge number of denoising tasks makes this computationally expensive to calculate the necessary per-task loss or gradient. To address this challenge, we propose clustering the denoising tasks into small task clusters and applying MTL methods to them. Specifically, based on **(O2)**, we employ interval clustering to enforce temporal proximity among denoising tasks within clusters. We show that interval clustering can be solved using dynamic programming, utilizing signal-to-noise ratio, timestep, and task affinity for clustering objectives. Through this, our approach addresses the issue of negative transfer in diffusion models by allowing for efficient computation of MTL methods. We validate the efficacy of proposed clustering and its integration with MTL methods through various experiments, demonstrating 1) improved generation quality and 2) faster training convergence of diffusion models. Our project page is available at https://gohyojun15.github.io/ANT_diffusion/.

## 1 Introduction

Diffusion-based generative models [20, 66, 71] have accomplished remarkable achievements in various generative tasks, including image [8], video [21, 23], 3D shape [44, 54], and text generation [38]. In particular, they have shown excellent performance and flexibility in a wide range of image generation settings, including unconditional [28, 47], class-conditional [22], and text-conditional image generation [1, 48, 55]. Consequently, improving diffusion models has garnered significant interest.

The framework of diffusion models [20, 66, 71] comprises gradually corrupting the data towards a given noise distribution and its subsequent reverse process. A model is optimized by minimizing the weighted sum of denoising score-matching losses across various noise levels [20, 69] for learning the reverse process. This can be interpreted as diffusion training aiming to train a single shared model to

---

*Co-first author   [1,2,4,5]Work done while at Riiid   †Corresponding author

denoising its input across various noise levels. Therefore, diffusion training is inherently multi-task learning (MTL) in nature, where *each noise level* represents *a distinct denoising task*.

However, analyzing and improving diffusion models from an MTL perspective remains under-explored. In particular, sharing one model between tasks may lead to competition between conflicting tasks, resulting in a phenomenon known as *negative transfer* [24, 25, 57, 78], leading to poorer performance compared to learning individual tasks with separate models. *Negative transfer* has been a critical issue in MTL research, and related works have demonstrated that the performance of multi-task models can be improved by remediating *negative transfer* [24, 25, 57, 78, 83]. Considering these, we argue that *negative transfer* should be investigated in diffusion models, and if present, addressing it is a potential direction for improving diffusion models.

In this paper, we characterize how *multi-task* diffusion model is, and whether there exists *negative transfer* in denoising tasks. In particular, **(O1)** we first observe that task affinity [12, 78] between two denoising tasks is negatively correlated with the difference in noise levels, indicating that they may be less conflict as the noise levels become more similar [78]. This suggests that adjacent denoising tasks should be considered more harmonious tasks than non-adjacent tasks in terms of noise levels.

Next, **(O2)** we observe the presence of *negative transfer* from diffusion model training. During sampling within a specific timestep interval, utilizing a model trained exclusively on denoising tasks within that interval generates higher-quality samples compared to a model trained on all denoising tasks simultaneously. This finding implies that simultaneously learning all denoising tasks can cause degraded denoising within a specific time interval, indicating the occurrence of *negative transfer*.

Based on these observations, we focus on improving diffusion models by addressing *negative transfer*. To achieve this, we first propose to leverage the existing multi-task learning techniques, such as dealing with issues of conflicting gradients [5, 83], differences in gradient magnitudes [42, 46, 64], and imbalanced loss scales [4, 16, 29]. However, unlike previous MTL studies that typically focused on small sets of tasks, the presence of a large number of denoising tasks ($\approx$ thousands) in diffusion models makes it computationally expensive since MTL methods generally require calculating per-task loss or gradient in each iteration [4, 5, 16, 24, 29, 42, 46, 64, 78, 83].

To address this, we propose a strategy that first clusters the entire denoising tasks and then applies multi-task learning methods to the resulting clusters. Specifically, inspired by **(O1)**, we formulate the interval clustering problem which groups denoising tasks by pairwise disjoint timestep intervals. Based on the interval clustering, we propose timesteps, signal-to-noise ratios, and task affinity score-based interval clustering and show that these can be clustered by dynamic programming as [2, 76, 49]. Through our strategy, we can address the issue of *negative transfer* in diffusion models by allowing for efficient computation of multi-task learning methods.

We evaluated our proposed methods through extensive experiments on widely-recognized datasets: FFHQ [27], CelebA-HQ [26], and ImageNet [7]. For a comprehensive analysis, we employed various models, including Ablated Diffusion Model (ADM) [8], Latent Diffusion Model (LDM) [56], and Diffusion Transformer (DiT) [52]. These models represent diverse diffusion architectures spanning pixel-space, latent-space, and transformer-based paradigms. Our results underscore a significant enhancement in image generation quality, attributed to a marked reduction in *negative transfer*. This affirms the merits of our clustering proposition and its synergistic integration with MTL techniques.

## 2 Related Work

**Diffusion Models** Diffusion models [20, 66, 71] are a family of generative models that generate samples from noise via a learned denoising process. Diffusion models beat other likelihood-based models, such as autoregressive models [62, 75], flow models [9, 10], and variational autoencoders [32] in terms of sample quality, and sometimes outperform GANs [14] in certain cases [8]. Moreover, pre-trained diffusion models can be easily applied to downstream image synthesis tasks such as image editing [30, 45] and plug-and-play generation [13, 15]. From these advantages, several works have applied diffusion models for various domains [3, 23, 38, 44, 54] and large-scale models [48, 56, 58].

Several studies have focused on improving diffusion models in various aspects, such as architecture [1, 8, 28, 52, 82], sampling speed [33, 60, 67], and training objectives [6, 17, 31, 70, 74]. Among these, the most closely related studies are improving training objectives, as we aim to enhance optimization between denoising tasks from the perspective of multi-task learning (MTL). Several works [31, 70, 74]

redesign training objectives to improve likelihood estimation. However, these objectives may lead to sample quality degradation and training instability and require additional techniques such as importance sampling [70, 74] and sophisticated parameterization [31] to be successfully applied. On the other hand, P2 [6] proposes a weighted training objective that prioritizes denoising tasks for certain noise levels, where the model is expected to learn perceptually rich features. Similar to P2, we aim to improve the sample quality of diffusion models from an MTL perspective, and we will show that our method is also beneficial to P2.

As a concurrent work, MinSNR [17] shares a common insight with us that diffusion training is essentially multi-task learning. However, their observation lacks a direct connection to *negative transfer* in terms of sample quality. They address the instability and inefficiency of multi-task learning optimization in diffusion models, mainly due to a large number of denoising tasks. In contrast, our work delves deeper into exploring *negative transfer* and task affinity, and we propose the application of MTL methods through task clustering to overcome the identified challenges in MinSNR.

**Multi-Task Learning** Multi-Task Learning (MTL) is an approach that trains a single model to perform multiple tasks simultaneously [57]. Although sharing parameters between tasks can reduce the overall number of parameters, it may also result in a *negative transfer*, causing performance degradation because of conflicting tasks during training procedure [24, 25, 57, 78].

Prior works have tracked down three causes of *negative transfer*: (1) *conflicting gradient*, (2) *the difference in gradient magnitude*, and (3) *imbalanced loss scale*. First, *Conflicting gradients* among different tasks may negate each other, resulting in poorer updates for a subset of, or even for all tasks. PCgrad [83] and Graddrop [5] mitigate this by projecting conflicting parts of gradients and dropping elements of gradients based on the degree of conflict, respectively. Second, tasks with larger gradients may dominate tasks with smaller gradients due to *differences in gradient magnitude* across tasks. Different optimization schemes have been proposed to equalize gradient magnitudes, including MGDA-UB [64], IMTL-G [42], and NashMTL [46]. Similarly, *imbalanced loss scales* may cause tasks with smaller losses to be dominated by those with larger losses. To balance task losses, uncertainty [29], task difficulty [16], and gradient norm [4] is exploited.

Adapting MTL methods and *negative transfer* formulation to diffusion models is challenging since these techniques are typically designed for scenarios with a small number of tasks and easily measurable individual task performance. Our goal is to address this challenge and demonstrate that observing *negative transfer* in diffusion models and mitigating it can improve them.

## 3 Preliminaries and Observation

We first provide the necessary background information on diffusion models and their multi-task nature. Next, we conduct analyses that yield two important observations: **(O1)** task affinity between two tasks is negatively correlated with the difference in noise levels, and **(O2)** *negative transfer* indeed exists in diffusion training, i.e., the model is overburdened with different, potentially conflicting tasks.

### 3.1 Preliminaries

Diffusion model [20, 66, 71] consists of two processes: a forward process and a reverse process. The forward process $q$ gradually injects noise into a datapoint $x_0$ to obtain noisy latents $\{x_1, \ldots, x_T\}$ as:

$$q(x_t|x_0) = \mathcal{N}(x_t|a_t x_0, \sigma_t^2 I), \quad q(x_t|x_s) = \mathcal{N}(x_t|\alpha_{t|s} x_s, (\sigma_t^2 - \alpha_{t|s}^2 \sigma_s^2) I), \quad 1 \leq s < t \leq T \quad (1)$$

where $\alpha_t, \sigma_t$ characterize the signal-to-noise ratio $\text{SNR}(t) = \alpha_t^2/\sigma_t^2$, and $\alpha_{t|s} = \alpha_t/\alpha_s$. Here, $\text{SNR}(t)$ decreases in $t$, such that by the designated final timestep $t = T$, $q(x_T) \approx \mathcal{N}(0, I)$.

The reverse process is a parameterized model trained to restore the original data from data corrupted during the forward process. The widely adopted training scheme uses a simple noise-prediction objective [8, 20, 34, 56, 59] that trains the model to predict the noise component $\epsilon$ of the latent $x_t = \alpha_t x_0 + \sigma\epsilon, \epsilon \sim \mathcal{N}(0, I)$. More formally, the objective is as follows:

$$L_{simple} = \mathbb{E}_{t,x_0,\epsilon}[L_t], \qquad \text{where } L_t = ||\epsilon - \epsilon_\theta(x_t, t)||_2^2. \quad (2)$$

Let us denote by $\mathcal{D}^t$ the denoising task at timestep $t$ trained by minimizing the loss $L_t$ (Eq. 2). Then, since a diffusion model jointly learns multiple denoising tasks $\{D_t\}_{t=1,\ldots,T}$ using a single shared model $\epsilon_\theta$, it can be regarded as a multi-task learner. Also, we denote by $\mathcal{D}^{[t_1,t_2]}$ the set of tasks $\{\mathcal{D}^{t_1}, \mathcal{D}^{t_1+1}, \ldots, \mathcal{D}^{t_2}\}$ henceforth.

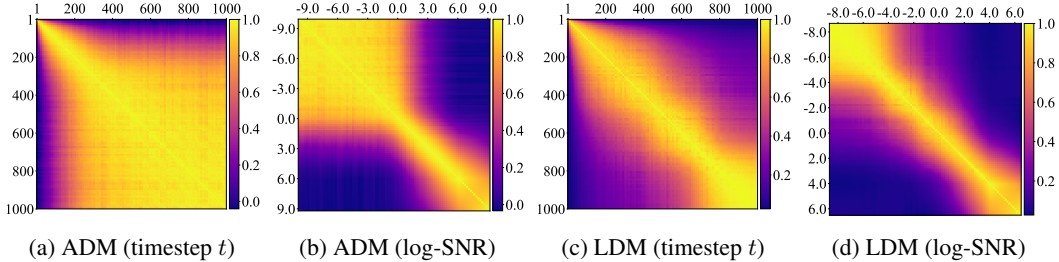

| (a) ADM (timestep $t$) | (b) ADM (log-SNR) | (c) LDM (timestep $t$) | (d) LDM (log-SNR) |

Figure 1: Task affinity scores plotted against timestep and log-SNR axes in ADM and LDM. As the timestep and SNR differences decrease, task affinity increases, implying more aligned gradient directions between denoising tasks and reduced negative impact on their joint training.

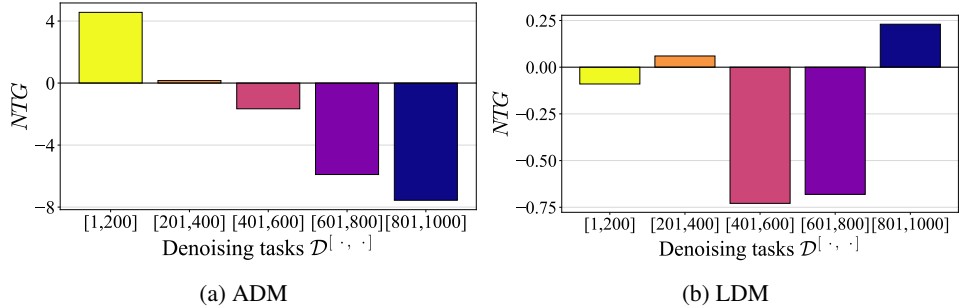

| (a) ADM | (b) LDM |

Figure 2: Negative transfer gap ($NTG$) with FID score of ADM and LDM for denoising tasks $\mathcal{D}^{[\cdot,\cdot]}$. If $NTG$ is negative, $\mathcal{D}^{[\cdot,\cdot]}$-trained model outperforms the entire denoising tasks-trained model in terms of denoising latent $\{\mathbf{x}_t\}_{t\in[\cdot,\cdot]}$, showing the occurrence of negative transfer. Negative transfer occurs in both ADM and LDM.

## 3.2 Observation

By considering diffusion training as a form of multi-task learning, we can analyze how the diffusion model learns the denoising task. We experimentally analyze diffusion models with two concepts in multi-task learning: 1) Task affinity [72, 12]: measuring which combinations of denoising tasks may yield a more positive impact on performance. 2) Negative transfer [68, 24, 25, 57, 78, 83]: degradation in denoising tasks caused by multi-task learning. We use a lightweight ADM [8] used in [6] and LDM [56] with FFHQ 256×256 dataset [27] for analyze diffusion models trained on both pixel and latent space.

**(O1) Task Affinity Analysis** We first analyze how the denoising tasks $\mathcal{D}^{[1,T]}$ relate to each other by measuring task affinities [72, 12]. In particular, we adopt the gradient direction-based task affinity score [78]: for two given tasks $\mathcal{D}^i$ and $\mathcal{D}^j$, we calculate the pairwise cosine similarity between gradients from each task loss, i.e., $\nabla_\theta L_i$ and $\nabla_\theta L_j$, then average the similarities across training iterations. Task affinity score assumes that cooperative (conflicting) tasks produce similar (conflicting) gradient directions, and it has been to correlate with the MTL model's overall performance [78]. Although there have been attempts to divide diffusion model phases using signal-to-noise ratio [6] and a trace of covariance of training targets [81], we are the first to provide an explicit and fine-grained analysis of task affinities among denoising tasks.

In Fig. 1, we visualize the task affinity scores among denoising tasks, for both ADM and LDM, with both timestep and log-SNR as axes. As can be seen in Fig. 1, task affinity between two tasks $\mathcal{D}^i, \mathcal{D}^j$ is high for neighboring tasks, i.e., $i \approx j$, and decreases smoothly as the difference in SNRs (or timesteps) increases. This suggests that tasks sharing temporal/noise-level proximity can be cooperatively learned without significant conflict. Also, this result hints at the possibility that denoising tasks for vastly different SNRs (distant in timesteps) may potentially be conflicting.

**(O2) Negative Transfer Analysis** Next, we show that there exist negative transfers among different denoising tasks $\mathcal{D}^{[1,T]}$. Negative transfer refers to a multi-task learner's performance degradation due to task conflicts, and it can be identified by observing the performance gap between a multi-task learner and specific-task learners. For ease of observation, we group up tasks by intervals, based on

the observation **(O1)** that more neighboring tasks in timesteps have higher task affinity. Specifically, we investigate whether the task group $\mathcal{D}^{[t_1,t_2]}$ suffers negative impacts from the remaining tasks.

To quantify the negative transfer, we follow the procedure: First, we generate samples $\{\tilde{\mathbf{x}}_0\}$ using a model trained on all denoising tasks $\mathcal{D}^{[1,T]}$. Next, we repeat the same sampling procedure, except we replace the model with a model trained on $\mathcal{D}^{[t_1,t_2]}$ for the latent $\{\mathbf{x}_t\}_{t\in[t_1,t_2]}$; We denote the resulting samples by $\{\tilde{\mathbf{x}}_0^{[t_1,t_2]}\}$. If $\{\tilde{\mathbf{x}}_0^{[t_1,t_2]}\}$ exhibits superior quality compared to $\{\tilde{\mathbf{x}}_0\}$, it indicates that the model trained solely on $\mathcal{D}^{[t_1,t_2]}$ performs better in denoising the latent $\{\mathbf{x}_t\}_{t\in[t_1,t_2]}$ than the model trained on the entire denoising task. This suggests that $\mathcal{D}^{[t_1,t_2]}$ suffers from negative transfer by learning other tasks. More formally, given a performance metric $P$, FID [18] in this paper, we define the negative transfer gap:

$$NTG(\mathcal{D}^{[t_1,t_2]}) := P(\{\tilde{\mathbf{x}}_0^{[t_1,t_2]}\}) - P(\{\tilde{\mathbf{x}}_0\}), \tag{3}$$

where $NTG < 0$ indicates that negative transfer occurs. The relationship between the negative transfer gap in previous literature and our negative transfer gap is described in Appendix A.

We visualize the negative transfers among denoising tasks for both lightweight ADM [6, 8] and LDM [56] in Fig. 2. The results indicate that negative transfer occurs in three out of the five considered task groups for both models. Notably, negative transfers often have a significant impact, such as a 7.56 increase in FID for ADM in the worst case. Therefore, we hypothesize that there is room for improving the performance of diffusion models by mitigating negative transfer, which motivates us to leverage well-designed MTL methods for diffusion training.

## 4 Methodology

In Section 3.2, we make two observations: **(O1)** Denoising tasks with a larger difference in $t$ and $\text{SNR}(t)$ exhibit lower task affinity, **(O2)** Negative transfer occurs in diffusion training. Inspired by these observations, we aim to remediate the negative transfer in diffusion by leveraging MTL methods. Although MTL methods are reported effective when there are only a few tasks, they are impractical for diffusion models with a large number of denoising tasks since they require computing per-task gradients or loss at each iteration. In this section, to deal with challenges, we propose a strategy that first groups the denoising tasks as task clusters and then applies the multi-task learning methods by regarding each task cluster as one distinct task.

### 4.1 Interval Clustering

Here, we first introduce a scheme that groups all denoising tasks $\mathcal{D}^{[1,T]}$ into a small number of task clusters. This is a necessary step for applying well-established MTL methods, for they usually involve computationally expensive subroutines such as computing per-task gradients or loss in each training iteration. Our key idea is to enforce temporal proximity of denoising tasks within task clusters, given our observation **(O1)** that task affinity is higher for tasks closer in timesteps. Therefore, we assign tasks in pairwise disjoint time intervals.

To obtain the disjoint time intervals, we leverage an interval clustering algorithm [2, 49] that optimizes for various clustering costs. In our case, interval clustering assigns diffusion timesteps $\mathcal{X} = \{1, \ldots, T\}$ to $k$ contiguous intervals $I_1, \ldots, I_k$, with $\coprod_{i=1}^k I_i \cap \mathcal{X} = \mathcal{X}$, where $\coprod$ denotes disjoint union. Let $I_i = [l_i, r_i]$, $l_i \le r_i$ for $i = 1, \ldots, k$, then we have $l_1 = 1$, and $r_i = l_{i+1} - 1$ ($i < k$ and $r_k = T$). The interval clustering problem is defined as:

$$\min_{l_1=1<l_2<\ldots<l_k} \sum_{i=1}^k L_{cluster}(I_i \cap \mathcal{X}), \tag{4}$$

where $L_{cluster}$ denotes the cluster cost.

Generally, it is known that an interval clustering problem of $n$ data points with $k$ intervals can be solved via dynamic programming in $O(n^2 k \omega(n))$ [49], where $\omega(n)$ is the time required to calculate the one-cluster cost for $L_{cluster}(\mathcal{X})$. If the size of each cluster is too small, it is challenging to learn the corresponding task cluster, so we add constraints on the cluster size for dynamic programming. More details regarding the dynamic programming algorithm can be found in Appendix G.

It remains to design the clustering cost function $L_{cluster}$ to optimize for. We present three clustering cost functions: timestep-based, SNR-based, and gradient-based.

**1. Timestep-based Clustering Cost**  Intuitively, one simple clustering cost is based on timesteps. We use the absolute timestep difference for the clustering objective by setting $L_{cluster}(I_i \cap \mathcal{X}) = \sum_{t=l_i}^{r_i} ||t_{center}^i - t||_1^1$ in Eq. 4 where $t_{center}^i$ denotes the center of interval $I_i$. The resulting intervals divide up the timesteps into $k$ uniform intervals.

**2. SNR-based Clustering Cost**  Another useful metric to characterize a denoising task is its signal-to-noise ratio (SNR). Indeed, it has been previously observed that a denoising task encounters perceptually different noisy inputs depending on its SNR [6]. Also, we already observed that denoising tasks with similar SNRs show high task affinity scores (see Section 3.2). Based on this, we use the absolute log-SNR difference for clustering cost. We define the clustering cost as $L_{cluster}(I_i \cap \mathcal{X}) = \sum_{t=l_i}^{r_i} ||\log \mathrm{SNR}(t_{center}^i) - \log \mathrm{SNR}(t)||_1^1$.

**3. Gradient-based Clustering Cost**  Finally, we consider the gradient direction-based task affinity scores (see Section 3.2 for a definition) for clustering cost. Task affinity scores have been used as a metric to group cooperative tasks [78]. Based on a similar intuition, we design a clustering cost as follows: $L_{cluster}(I_i \cap \mathcal{X}) = - \sum_{t=l_i}^{r_i} \mathrm{TAS}(t_{center}^i, t)$ where $\mathrm{TAS}(\cdot)$ is the gradient-based task affinity score. While leveraging more fine-grained information regarding task affinities, this cost function requires computing and storing gradients throughout training.

## 4.2  Incorporating MTL Methods into Diffusion Model Training

After dividing the denoising tasks into task clusters via interval clustering, we apply multi-task learning methods to the resulting task clusters. As mentioned in Section 2, previous multi-task learning works have tracked down the following causes for negative transfer: (1) *conflicting gradient*, (2) *difference in gradient magnitude*, and (3) *imbalanced loss scale*. In this work, we leverage one representative method that tackles each of the causes mentioned above, namely, (1) PCgrad [83], (2) NashMTL [46], and (3) Uncertainty Weighting [29].

For each training step in diffusion modeling, we compute the noise prediction loss $L^l$ for the $l$-th data within the minibatch. As shown in Eq 2, calculating $L^l$ involves sampling the timestep $t^l$, in which case $L^l$ is a loss incurred on the denoising task $\mathcal{D}^{t^l}$. We may then assign $L^i$ to the appropriate task cluster by considering the corresponding timestep. Subsequently, we may group up the losses as $\{L_{I_i}\}_{i=1,\ldots,k}$, where $L_{I_i}$ is the loss for the $i$-th task cluster. (More details in Appendix C)

**1. PCgrad** [83] In each iteration, PCgrad projects the gradient of a task onto the normal plane of the gradient of another task when there is a conflict between their gradients. Specifically, PCgrad first calculates the per-interval gradient $\nabla_\theta L_{I_i}$. Then, if the other interval gradient $\nabla_\theta L_{I_j}$ for $i \neq j$ has negative cosine similarity with $\nabla_\theta L_{I_i}$, it projects $\nabla_\theta L_{I_i}$ onto the normal plane of $\nabla_\theta L_{I_j}$. PCgrad repeats this process with all of the other interval gradients for all interval gradients, resulting in a projected gradient per interval. Finally, model parameters are updated with the summation of projected gradients.

**2. NashMTL** [46] In NashMTL, the aggregation of per-task gradients is treated as a bargaining game. It aims to update model parameters with weighted summed gradients $\Delta\theta = \sum_{i=i}^{k} \alpha_i \nabla_\theta L_{I_i}$ by obtaining the Nash bargaining solution to determine $\alpha_i$, where $\Delta\theta$ is in the ball of radius $\epsilon$ centered zero, $B_\epsilon$. They define the utility function for each player as $u_i = \langle \nabla_\theta L_{I_i}, \Delta\theta \rangle$, then the unique Nash bargaining solution can be obtained by $\arg\max_{\Delta\theta \in B_\epsilon} \sum_i \log(u_i)$. By denoting $G$ as matrix whose columns contain the gradients $\nabla_\theta L_{I_i}$, $\alpha \in \mathbb{R}_+^k$ is the solution to $G^\mathsf{T} G\alpha = 1/\alpha$ where $1/\alpha$ is the element-wise reciprocal. To avoid the optimization to obtain $\alpha$ for each iteration, they update $\alpha$ once every few iterations.

**3. Uncertainty Weighting (UW)** [29] UW uses task-dependent (homoscedastic) uncertainty to weight task cluster losses. By utilizing observation noise parameter $\sigma_i$ for $i$-th task clusters, the total loss function is $\sum_i L_{I_i}/\sigma_i^2 + \log(\sigma_i)$. As the noise parameter for the $i$-th task clusters loss $\sigma_i$ increases, the weight of $L_{I_i}$ decreases, and vice versa. The $\sigma_i$ is discouraged from increasing too much by regularizing with $\log(\sigma_i)$.

# 5  Experiments

In this section, we demonstrate the efficacy of our proposed method by addressing the negative transfer issue in diffusion training. First, we provide the comparative evaluation in Section 5.1, where

Table 1: Quantitative comparison to vanilla training (Vanilla) on the unconditional generation. Integration of MTL methods using interval clustering consistently improves FID scores and generally enhances precision compared to vanilla training.

| Model | Clustering | Method | Dataset | | | | | |
|---|---|---|---|---|---|---|---|---|
| | | | FFHQ [27] | | | CelebA-HQ [26] | | |
| | | | FID ($\downarrow$) | Precision ($\uparrow$) | Recall ($\uparrow$) | FID ($\downarrow$) | Precision ($\uparrow$) | Recall ($\uparrow$) |
| ADM [8, 6] | | Vanilla | 24.95 | 0.5427 | 0.3996 | 22.27 | 0.5651 | 0.4328 |
| | Timestep | PCgrad [83] | 22.29 | 0.5566 | 0.4027 | 21.31 | 0.5610 | 0.4238 |
| | | NashMTL [46] | 21.45 | 0.5510 | **0.4193** | 20.58 | 0.5724 | 0.4303 |
| | | UW [29] | 20.78 | 0.5995 | 0.3881 | **17.74** | **0.6323** | 0.4023 |
| | SNR | PCgrad [83] | 20.60 | 0.5743 | 0.4026 | 20.47 | 0.5608 | 0.4298 |
| | | NashMTL [46] | 23.09 | 0.5581 | 0.3971 | 20.11 | 0.5733 | **0.4388** |
| | | UW [29] | **20.19** | **0.6297** | 0.3635 | 18.54 | 0.6060 | 0.4092 |
| | Gradient | PCgrad [83] | 23.07 | 0.5526 | 0.3962 | 20.43 | 0.5777 | 0.4348 |
| | | NashMTL [46] | 22.36 | 0.5507 | 0.4126 | 21.18 | 0.5682 | 0.4369 |
| | | UW [29] | 21.38 | 0.5961 | 0.3685 | 18.23 | 0.6011 | 0.4130 |
| LDM [56] | | Vanila | 10.56 | 0.7198 | 0.4766 | 10.61 | 0.7049 | 0.4732 |
| | Timestep | PCgrad [83] | 9.599 | 0.7349 | 0.4845 | 9.817 | 0.7076 | 0.4951 |
| | | NashMTL [46] | 9.400 | 0.7296 | 0.4877 | 9.247 | 0.7119 | 0.4945 |
| | | UW [29] | 9.386 | 0.7489 | 0.4811 | 9.220 | 0.7181 | 0.4939 |
| | SNR | PCgrad [83] | 9.715 | 0.7262 | 0.4889 | 9.498 | 0.7071 | 0.5024 |
| | | NashMTL [46] | 10.33 | 0.7242 | 0.4710 | 9.429 | 0.7062 | 0.4883 |
| | | UW [29] | 9.734 | 0.7494 | 0.4797 | **9.030** | **0.7202** | 0.4938 |
| | Gradient | PCgrad [83] | **9.189** | 0.7359 | 0.4904 | 10.31 | 0.6954 | 0.4927 |
| | | NashMTL [46] | 9.294 | 0.7234 | **0.4962** | 9.740 | 0.7051 | **0.5067** |
| | | UW [29] | 9.439 | **0.7499** | 0.4855 | 9.414 | 0.7199 | 0.4952 |

our method can boost the quality of generated samples significantly. Next, we compare previous loss weighting methods for diffusion models to UW with interval clustering in Section 5.2, verifying our superior effectiveness to existing methods. Then, we analyze the behavior of adopted MTL methods, which serve to explain the effectiveness of our method in Section 5.3. Finally, we demonstrate that our method can be readily combined with more sophisticated training objectives to boost performance even further in Section 5.4. Extensive information on all our experiments can be found in Appendix E.

## 5.1 Comparative Evaluation

**Experimental Setup** Here, we demonstrate that incorporating MTL methods into diffusion training improves the performance of diffusion models. For comparison, we consider unconditional and class-conditional image generation. For unconditional image generation, we used FFHQ [27] and CelebA-HQ [26] datasets, where all images were resized to $256 \times 256$. For class-conditional image generation experiments, we employed the ImageNet dataset [7], also resized to $256 \times 256$ resolution.

For architecture, we adopt widely recognized architectures for image generation. Specifically, we use the lightweight ADM [6, 8] and LDM [56] for unconditional image generation, while employing DiT-S/2 [52] with classifier-free guidance [19] for class-conditional image generation. We train the model using our method: We consider every possible pair of (1) interval clustering (timestep-, SNR-, and gradient-based) and (2) MTL method (PCgrad, NashMTL, and Uncertainty Weighting (UW)), and report the results. We used $k = 5$ in interval clustering throughout experiments.

For evaluation metrics, we use FID [18] and precision [36] for measuring sample quality, and recall [36] for assessing sample diversity and distribution coverage. IS [61] is additionally used for the evaluation metric in the class-conditional image generation setting. Finally, for sample generation, we use DDIM [67] sampler with 50 steps for unconditional generation and DDPM 250 steps for class conditional generation, and all evaluation metrics are calculated using 10k generated samples.

**Comparison in Unconditional Generation** As seen in Table 1 our method significantly improves performance upon conventionally trained diffusion models (denoted vanilla in the table). In particular, there is an improvement in FID in all cases, and an improvement in precision scores in all but two cases, which highlights the efficacy of our method. Also, given strong results for both pixel- and latent-space models, we can reasonably infer that our method is generally applicable.

We also observe the distinct characteristics of each multi-task learning method considered. Uncertainty Weighting tends to achieve higher improvements in sample quality compared to PCgrad and NashMTL. Indeed, UW achieves superior FID and Precision for ADM, while excelling in Precision for LDM.

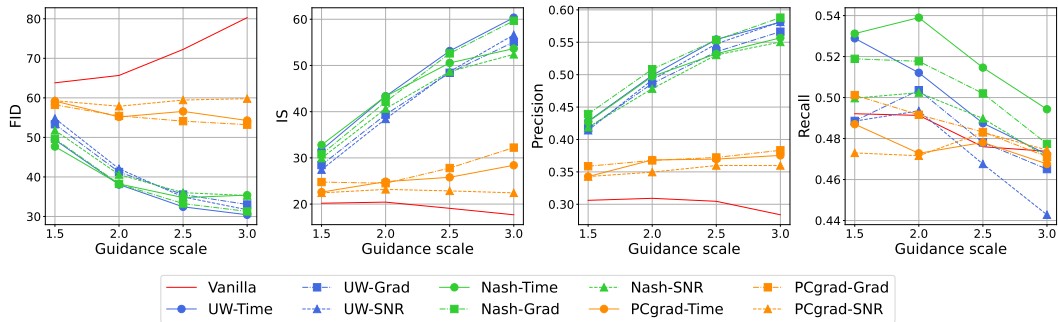

Figure 3: Quantitative comparison to vanilla training (Vanilla) on ImageNet 256×256 dataset with DiT-S/2 architecture and classifier-free guidance. Integration of MTL methods using interval clustering consistently improves FID, IS, and Precision compared to vanilla training.

Table 2: Comparison between MinSNR and ANT-UW. DiT-L/2 is trained on ImageNet.

| Method | FID | IS | Precision | Recall |
|--------|------|--------|-----------|--------|
| Vanilla | 12.59 | 134.60 | 0.73 | 0.49 |
| MinSNR | 9.58 | 179.98 | 0.78 | **0.47** |
| ANT-UW | **6.17** | **203.45** | **0.82** | **0.47** |

Table 3: GPU memory usage and runtime comparison on FFHQ dataset in LDM architecture.

| Method | GPU memory usage (GB) | # Iterations / Sec |
|--------|-----------------------|--------------------|
| Vanilla | 34.126 | **2.108** |
| PCgrad | **28.160** | 1.523 |
| NashMTL | 38.914 | 2.011 |
| UW | 34.350 | 2.103 |

However, UW sacrifices distribution coverage in exchange for sample quality, resulting in lower Recall compared to other methods. Meanwhile, NashMTL scores higher in recall and lower in precision compared to other methods, suggesting it has better distribution coverage while sacrificing sample quality. Finally, PCgrad tends to show a balanced performance in terms of precision and recall. We further look into behaviors of different MTL methods in Section 5.3.

Due to space constraints, we provide a comprehensive collection of generated samples in Appendix F. In summary, diffusion models trained with our method produce more realistic and high-fidelity images compared to conventionally trained diffusion models.

**Comparison in Class-Conditional Generation** We illustrate the results of quantitative comparison on class-conditional generation in Fig. 3. The results show that our methods outperform vanilla training in FID, IS, and Precision. In particular, UW and Nash-MTL significantly boost these metrics, showing superior improvement in generation quality. These results further support the generalizability of MTL methods through the interval clustering on class-conditional generation and the transformer-based diffusion model.

## 5.2 Comparison to Loss Weighting Methods

Since UW is a loss weighting method, validating the superiority of UW with interval clustering compared to previous loss weighting methods such as P2 [6] and MinSNR [17] highlights the effectiveness of our method. We name UW by incorporating interval clustering as Addressing Negative Transfer (ANT)-UW. We trained DiT-L/2 with MinSNR and UW with $k = 5$ on the ImageNet across 400K iterations, using a batch size of 240. All methods are trained by AdamW optimizer [43] with a learning rate of $1e - 4$. Table 2 shows that ANT-UW dramatically outperforms MinSNR, emphasizing the effectiveness of our method. An essential note is that the computational cost of ANT-UW remains remarkably similar to vanilla training as shown in Section 5.3, ensuring that our enhanced performance does not come at the expense of computational efficiency. Additionally, we refer to the results in [50], showing that our ANT-UW outperforms P2 and MinSNR when DIT-L/2 is trained on the FFHQ dataset.

## 5.3 Analysis

To provide a better understanding of our method, we present various analysis results here. Specifically, we compare the memory and runtime of MTL methods, analyze the behavior of MTL methods adopted, provide a convergence analysis, and assess the extent to which negative transfer has been addressed.

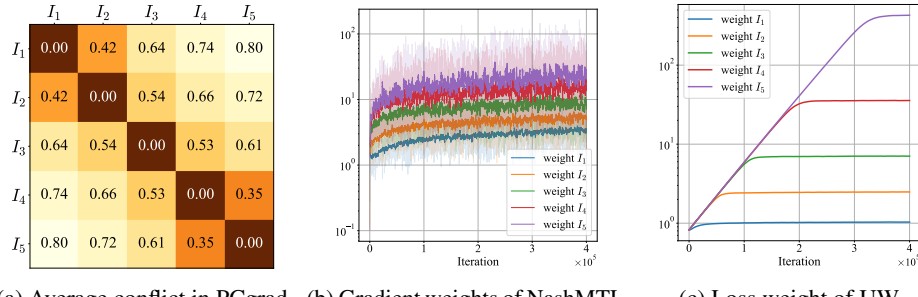

(a) Average conflict in PCgrad    (b) Gradient weights of NashMTL    (c) Loss weight of UW

Figure 4: Behavior of multi-task learning methods across training iterations. (a): With increasing timestep difference, gradient conflicts between task clusters become more frequent in PCgrad. (b) and (c): Both UW and NashMTL allocate higher weights to task clusters that handle noisier inputs.

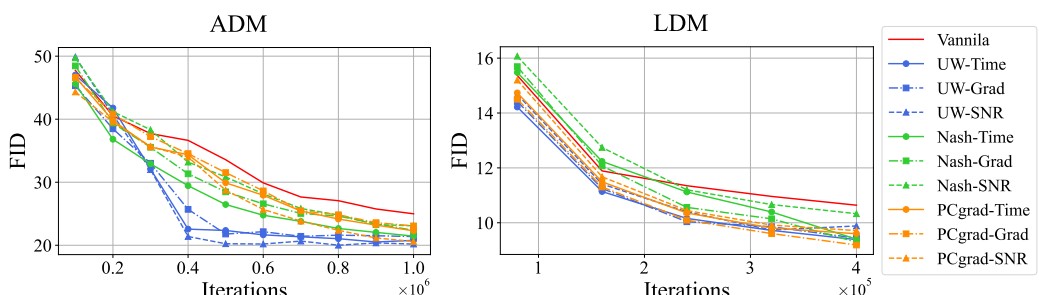

Figure 5: Convergence analysis on FFHQ dataset. Compared to baselines, all methods exhibit fast convergence and achieve good final performance.

**Memory and Runtime Comparison**  We first compared the memory usage and runtime between MTL methods and vanilla training for a deeper understanding of their cost. We conducted measurements of memory usage and runtime with $k = 5$ on the FFHQ dataset using the LDM architecture and timestep-based clustering, and the results are shown in Table 3. PCgrad has a slower speed of 1.523 iterations/second compared to vanilla training, but its GPU memory usage is lower due to the partitioning of minibatch samples. Meanwhile, NashMTL has a runtime of 2.011 iterations/second. Even though NashMTL uses more GPU memory, it has a better runtime than PCgrad because it computes per-interval gradients occasionally. Concurrently, UW shows similar runtime and GPU memory usage as vanilla training, which is attributed to its use of weighted loss and a single backpropagation process.

**Behavior of MTL Methods**    We analyze the behavior of different multi-task learning methods during training. For PCgrad, we calculate the average number of gradient conflicts between task clusters per iteration. For UW, we visualize the weights allocated to the task cluster losses over training iterations. Finally, for NashMTL, we visualize the weights allocated to per-task-cluster gradients over training iterations. We used LDM trained on FFHQ for our experiments. Although we only report results for time-based interval clustering for conciseness, we note that MTL methods exhibit similar behavior across different clustering methods. Results obtained using other clustering methods can be found in Appendix D.1.

The resulting visualizations are provided in Fig. 4. As depicted in Fig. 4a, the task pair that shows the most gradient conflicts is $I_1$ and $I_5$, namely, task clusters apart in timesteps. This result supports our hypothesis that temporally distant denoising tasks may be conflicting, and as seen in Section 5.1, PCgrad seems to mitigate this issue. Also, as depicted in Fig. 4b and 4b, both UW and NashMTL tend to allocate higher weights to task clusters that handle noisier inputs, namely, $I_4, I_5$. This result suggests that handling noisier inputs may be a difficult task that is underrepresented in conventional diffusion training.

**Faster Convergence**  In Fig. 5, we plot the trajectory of the FID score over training iterations, as observed while training on FFHQ. We can observe that all our methods enjoy faster convergence and better final performance compared to the conventionally trained model. Notably, for pixel space

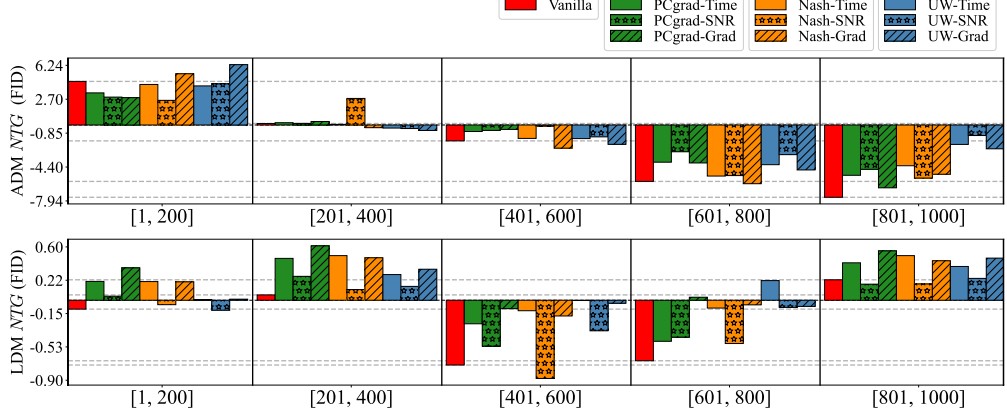

Figure 6: Negative transfer gap (NTG) comparison on the FFHQ dataset. Integration of MTL methods tends to improve the negative transfer gap. Methods that fail to improve NTG in areas where the baseline records low NTG tend to achieve lesser improvements in the baseline.

diffusion (ADM), UW converges extremely rapidly, while beating the vanilla method by a large margin. Overall, these results show that our method may not only make diffusion training more effective but also more efficient.

**Reduced Negative Transfer Gap** We now demonstrate that our proposed method indeed mitigates the negative transfer gap we observed in Section 3.2. We used the same procedure introduced in Section 3.2 to calculate the negative transfer gap for all methods considered, for the FFHQ dataset.

As shown in Fig. 6 our methods improve upon negative transfer gaps. Specifically, for tasks that exhibit severe negative transfer gaps in the baseline (e.g., [601, 800], [801, 1000] for ADM, and [401, 600], [601, 800] for LDM), our methods mitigate the negative transfer gap for most cases, even practically removing it in certain cases. Another interesting result to note is that models less effective in reducing negative transfer (NashMTL-SNR for LDM and PCgrad-Grad for ADM) indeed show worse FID scores, which supports our hypothesis that resolving negative transfer leads to performance gain. We also note that even the worst-performing methods still beat the vanilla model.

## 5.4 Combining MTL Methods with Sophisticated Training Objectives

Finally, we show that our method is readily applicable on top of more sophisticated training objectives proposed in the literature. Specifically, we train an LDM by applying both UW and PCgrad on top of the P2 objective [6] and evaluate the performance on the FFHQ dataset. We chose UW and PCgrad based on a previous finding that combining the two methods leads to performance gain [41]. Also, we chose the gradient-based clustering method due to its effectiveness for LDM on FFHQ. As seen in Table 4, when combined with P2, our method improves the FID from 7.21 to 5.84.

Table 4: Combining our method with P2 on the FFHQ dataset. DDIM 200-step sampler is used.

| Type | Method | FID-50k |
|------|--------|---------|
| GAN | PGan [63] | 3.39 |
| AR | VQGAN [11] | 9.6 |
| Diffusion (LDM) | D2C [65] | 13.04 |
| | Vanilla | 9.1 |
| | P2 | 7.21 |
| | P2 + Ours | **5.84** |

## 6 Conclusion

In this work, we studied the problem of better training diffusion models, with the distinction of reducing negative transfer between denoising tasks in a multi-task learning perspective. Our key contribution is to enable the application of existing multi-task learning techniques, such as PCgrad and NashMTL, that were challenging to implement due to the increasing computation costs associated with the number of tasks, by clustering the denoising tasks based on their various task affinity scores. Our experiments validated that the proposed method effectively mitigated negative transfer and improved image generation quality. Overall, our findings contribute to advancing diffusion models. Starting from our work, we believe that addressing and overcoming negative transfer can be the future direction to improve diffusion models.

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

# Appendix

## Contents

# A   Relation to Negative Transfer Gap in Previous Literature

Previous works on transfer and multi-task learning have explored measuring the negative transfer [25, 79, 77]. For the source task $\mathcal{T}_{src}$ and the target task $\mathcal{T}_{tgt}$, the negative transfer can be defined as the phenomenon that the source task *negatively* transfer to the target task. Denote the model trained on both source and target task as $\Theta(\mathcal{T}_{tgt}, \mathcal{T}_{src})$ and the model only trained on the target task as $\Theta(\mathcal{T}_{tgt})$. With performance measure $P$ for the model on $\mathcal{T}_{tgt}$, negative transfer can be quantified by utilizing negative transfer gap ($NTG$):

$$NTG(\mathcal{T}_{tgt}, \mathcal{T}_{src}) = P(\Theta(\mathcal{T}_{tgt})) - P(\Theta(\mathcal{T}_{tgt}, \mathcal{T}_{src})). \tag{5}$$

For $P$, higher is better, $NTG > 0$ indicates that negative transfer occurs, showing that additionally training on $\mathcal{T}_{src}$ negatively affects the learning of $\mathcal{T}_{tgt}$.

In our study of negative transfer in diffusion models, the target task involves denoising tasks within a specific timestep interval as $\mathcal{T}_{tgt} = \mathcal{D}^{[t_1, t_2]}$, while the source task comprises the remaining denoising tasks as $\mathcal{T}_{src} = \mathcal{D}^{[1,T]} \setminus \mathcal{D}^{[t_1, t_2]}$.

However, since a model trained only a subset of entire denoising tasks cannot generate samples properly, we cannot utilize the sample quality metrics (*e.g.* FID [18]) for $P$ to measure $P(\Theta(\mathcal{T}_{tgt}))$ in Eq. 5 for arbitrary timestep intervals. This is a different point from a typical MTL setting, where the performance of each task can be measured.

Alternatively, we redefine $NTG$ with the difference in sample quality resulting from denoising by different models, $\Theta(\mathcal{T}_{tgt})$ and $\Theta(\mathcal{T}_{tgt}, \mathcal{T}_{src})$, in the $[t_1, t_2]$ interval. During the sampling procedure with a model trained on entire denoising tasks, we use $\Theta(\mathcal{T}_{tgt})$ or $\Theta(\mathcal{T}_{tgt}, \mathcal{T}_{src})$ in $[t_1, t_2]$. Denote the resulting samples with $\Theta(\mathcal{T}_{tgt}, \mathcal{T}_{src})$ as $\{\tilde{x}_0\}$ and the resulting samples with $\Theta(\mathcal{T}_{tgt})$ as $\{\tilde{x}_0^{[t_1, t_2]}\}$. Then, by comparing the quality of these samples as Eq. 3, we can measure how much the denoising of $[t_1, t_2]$ degrades in terms of sampling quality.

Furthermore, the success of multi-expert denoisers in prior studies [37, 13, 1] suggests the potential existence of negative transfer. By distinctly separating parameters for denoising tasks, they might mitigate this negative transfer, leading to enhanced performance in their generation.

# B   Detailed Experimental Settings for Observational Study

In this section, we provide the details on experimental settings in Section 3. The training details and the architectures used are the same as those in Section 5. All experiments are conducted with a single A100 GPU and with FFHQ dataset [27].

For the pixel-space diffusion model, we use the lightweight ADM as same in [6]. It inherits the architecture of ADM [8], but it uses fewer base channels, fewer residual blocks, and a self-attention with a single resolution. Specifically, the model uses one residual block per resolution with 128 base channels and $16 \times 16$ self-attention with 64 head channels. A linear schedule with $T = 1000$ is used for diffusion scheduling. We referenced the training scripts in the official code[2] for implementation.

For the latent-space diffusion model, we use the LDM architecture as the same settings for FFHQ experiments in [56]. Specifically, an LDM-4-VQ encoder and decoder are used, in which the resolution of latent vectors is reduced by four times compared to the original images and has a vector quantization layer with 8092 codewords. The denoising model has 224 base channels with multipliers for each resolution as 1, 2, 3, 4 and has two residual blocks per resolution. Self-attention with 32 head channels is used for 32, 16, and 8 resolutions. For diffusion scheduling, the linear schedule with $T = 1000$ is used. We conducted experiments with the official code[3]. In general, we utilized the pre-trained weights provided by LDM. However, if our retraining results demonstrated superior performance, we reported them.

**Task Affinity Analysis**   To measure the task affinity score between denoising tasks, we first calculate $\nabla_\theta L_t$ for $t = 1, \ldots, T$ every 10K iterations during training. The gradient is calculated with 1000 samples in the training dataset. Then, the pairwise cosine similarity of the gradient is computed and

---

[2]https://github.com/jychoi118/P2-weighting
[3]https://github.com/CompVis/latent-diffusion

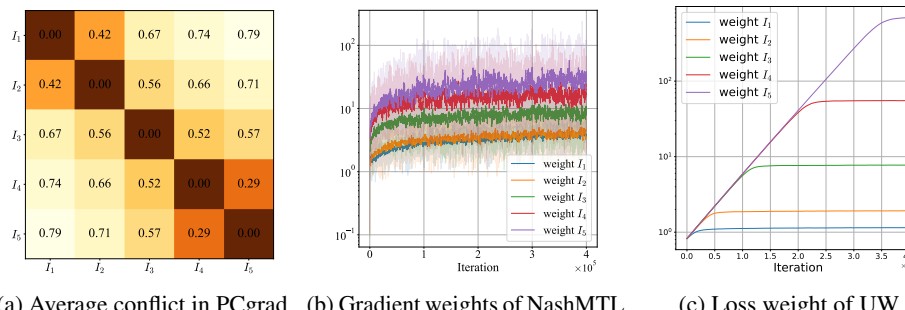

(a) Average conflict in PCgrad    (b) Gradient weights of NashMTL    (c) Loss weight of UW

Figure 7: Behavior of multi-task learning methods through SNR-based interval clustering across training iterations. A similar trend as in Fig. 3 is observed.

their cosine similarities calculated by every 10K iterations are averaged. Finally, we can plot the average cosine similarity against the timestep axis as in Fig. 1. For plotting them against the log-SNR axis, the values of the axis were adjusted, and the empty parts were filled with linear interpolation.

For ADM and LDM, the pairwise cosine similarity between gradients is calculated during 1M training iterations and 400K training iterations, respectively.

**Negative Transfer Analysis**    To calculate the negative transfer gap in Eq. 3, we need to additionally train the model on denoising tasks within specific timestep interval $[t_1, t_2]$. Since we plot five intervals [1, 200], [201, 400], [401, 600], [601, 800], and [801, 1000], we trained the model on denoising tasks for each interval. Each model is trained for 600K iterations in ADM and 300K iterations in LDM on the FFHQ dataset. For the model trained on entire denoising tasks, we used the trained model the same as in Section 5.1. ADM is trained on 1M iterations and LDM is trained on 400K iterations. All of these models are trained with the same batch size and learning rate as experiments in Section 5.1 (See Appendix E).

DDIM 50-step sampler [67] was used for the generation. FID is calculated with Clean-FID [51] by setting the entire 70K FFHQ dataset as reference images. Since the official code of Clean-FID[4] supports FID calculation with statistics from these reference images, we used it and reported FID with 10k generated images.

## C    Implementation Details for MTL methods

We describe how MTL methods are applied in Section 4.2. To be more self-contained, we hereby present implementation details for MTL methods. For the implementation of MTL methods, we used the official code of LibMTL [39][5]. NashMTL [46] supports practical speed-up by updating gradient weights $\alpha$ every few iterations, not every iteration. We utilize this by updating $\alpha$ every 25 training iterations.

## D    Additional Experimental Results

We present additional experimental results to supplement the empirical findings presented in Section 5. In Section D.1, we provide visualizations of the behavior of MTL methods with other clustering methods that were not covered in Section 5.3. Furthermore, we examine the impact of our hyperparameter, the number of clusters $k$, in Section D.2. To validate the effectiveness of interval clustering compared to other clustering methods, we present additional results in Section D.3. In Section D.4, we delve deeper into comparing the performance of stronger MTL baselines such as Linear Scalarization (LS) [80, 35] and Random Loss Weighting (RLW) [40] with our proposed approach.

---

[4] https://github.com/GaParmar/clean-fid
[5] https://github.com/median-research-group/LibMTL

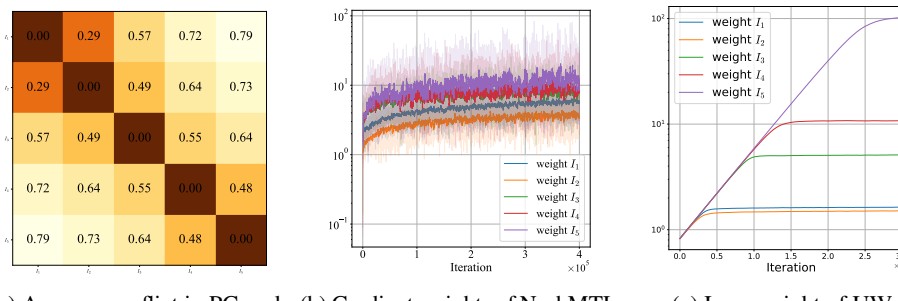

(a) Average conflict in PCgrad    (b) Gradient weights of NashMTL    (c) Loss weight of UW

Figure 8: Behavior of multi-task learning methods through gradient-based interval clustering across training iterations. A similar trend as in Fig. 3 is observed.

Table 5: FID-10K scores of the LDM trained using a combination of UW and PCgrad methods on the FFHQ dataset while varying the value of $k$. Notably, integrating MTL methods with two clusters significantly improves FID scores. Increasing $k$ from 2 to 5 also enhances FID scores, but further increasing $k$ from 5 to 8 shows similar results.

| Clustering | Vanilla | Number of clusters ($k$) | | |
| --- | --- | --- | --- | --- |
| | | $k = 2$ | $k = 5$ | $k = 8$ |
| Timestep | | 9.563 | 9.151 | 9.083 |
| SNR | 10.56 | 9.606 | 9.410 | 9.367 |
| Gradient | | 9.634 | **9.033** | 9.145 |

## D.1 Visualization for the Behavior of MTL Methods with Other Clustering Methods

Due to space constraints in our main paper, we were unable to include the behavior analysis of MTL methods for SNR-based and gradient-based interval clustering. However, we present these results in Fig. 7 and 8, which show similar trends to the observations depicted in Fig. 4. These findings suggest valuable insights into the behavior of MTL methods, regardless of the clustering objectives.

Firstly, we observed a notable increase in the occurrence of conflicting gradients as the timestep difference between tasks increased. This observation suggests that the temporal distance between denoising tasks plays a crucial role in determining the frequency of conflicting gradients.

Secondly, we noted that both loss and gradient balancing methods assign higher weights to task clusters with higher levels of noise. This finding indicates that these methods allocate more importance to the noisier tasks.

## D.2 Analysis: The Number of Interval Clusters

To understand the impacts of the number of clusters $k$, we conducted experiments by varying $k$ with 2, 5, and 8. We trained a model for timestep-based, SNR-based, and gradient-based clustering with each $k$, resulting in nine trained models. For MTL methods, we used combined methods with UW [29] and PCgrad [83] as in Section 5.4. All training configurations such as learning rate and training iterations are the same as in Section 5.1. We evaluate 10K generated samples from the DDIM 50-step sampler [67] for all methods with the FID score [51, 18].

Table 5 shows the results. Notably, we made an intriguing observation regarding the integration of MTL methods with only two clusters, which resulted in a noteworthy enhancement in FID scores. Additionally, we found that increasing the number of clusters, denoted as $k$, from 2 to 5 also exhibited a positive impact on improving FID scores. However, our findings indicated that further increasing $k$ from 5 to 8 did not yield significant improvements and resulted in similar outcomes. From these results, we conjecture that increasing the number of clusters to greater than five has no significant effect.

Table 7: The results of Random Loss Weighting (RLW) and Linear Scalarization (LS) on the FFHQ dataset in ADM architecture.

| Clustering | Method | FID | Precision | Recall |
|---|---|---|---|---|
| - | Vanilla | **24.95** | 0.5427 | **0.3996** |
| Timestep | RLW | 38.06 | 0.4634 | 0.3293 |
| | LS | 25.34 | **0.5443** | 0.3868 |
| SNR | RLW | 35.13 | 0.4675 | 0.3404 |
| | LS | 25.69 | 0.5369 | 0.3843 |
| Gradient | RLW | 36.19 | 0.4643 | 0.3392 |
| | LS | 26.12 | 0.5120 | 0.3878 |

### D.3 Comparison Interval Clustering with Task Grouping Method

To show the effectiveness of interval clustering methods for denoising task grouping in diffusion models, we compare high-order approximation (HOA)-based grouping methods [72, 12].

For grouping $N$-tasks in deep neural networks, the early attempt [72] established a two-stage procedure: (1) compute MTL performance gain for all task combinations and (2) search best groups for maximizing MTL performance gain across the groups. However, performing (1) requires huge computation since MTL performance gain should be measured for all $2^N - 1$ combinations. Therefore, they reduce computation by HOA, which utilizes MTL gains on only pairwise task combinations. Also, the HOA scheme is inherited by the following work, task affinity grouping [12], which uses their defined task affinity score instead of MTL gains. Different from these works, our interval clustering aims to group the tasks with interval constraints.

For a fair comparison, we use a pairwise gradient similarity averaged across training iterations between denoising tasks for the objective of HOA-based grouping and interval clustering. In this case, the HOA-based grouping becomes cosine similarity grouping used in [12], and interval clustering becomes gradient-based clustering in our method. However, for HOA-based grouping, a solution of brute force searching with branch-and-bound-like algorithm [72, 12] requires computational complexity of $O(2^N)$. It incurs enormous costs in diffusion with many denoising tasks. Therefore, we use a beam-search scheme in [68]. We set the number of clusters as 5 for both methods.

Table 6: Comparison interval clustering and high order approximation-based task grouping. DDIM-50 step sampler is used.

| Clustering | FID-10k |
|---|---|
| HOA | 9.873 |
| Interval | **9.033** |

We apply the combined method with UW [29] and PCgrad [83] as in Section 5.3 for the resulting clusters from both HOA-based grouping and interval clustering. We trained the model on the FFHQ dataset [27] and used LDM architecture [56]. All training configurations are the same as in Section 5.1. For evaluation metrics, we use FID and its configurations are the same as in Section 5.1.

Table 6 shows the results, indicating that the interval clustering outperforms HOA-based task grouping.

### D.4 Comparison to Random Loss Weighting and Linear Scalarization

Linear Scalarization (LS) [80, 35] and Random Loss Weighting (RLW) [40] can serve as strong baselines for MTL methods. Therefore, validating the superiority of our method compared to theirs can emphasize the necessity of applying sophisticated MTL methods such as UW, PCgrad, and NashMTL. Accordingly, we provide the results of comparative experiments for LS and RLW on the FFHQ dataset using ADM architecture in Table 7. We note that all experimental configuration is the same as in vanilla training in Section 5.1.

As shown in the results, LS achieves slightly worse performance than vanilla training, which suggests that simply re-framing the diffusion training task as an MTL task and applying LS is not enough. Also, RLW achieves much worse performance compared to vanilla training. It appears that the randomness introduced by loss weighting interferes with diffusion training. These results indicate that sophisticated MTL methods are indeed responsible for significant performance gain.

# E  Detailed Experimental Settings in Section 5

In this section, we describe the details of experimental settings in Section 5. For validating the effectiveness in both pixel-space and latent-space diffusion models in unconditional generation, we used ADM [8] and LDM [56] as same in our observational study (refer to details of architecture in Appendix B).

## E.1  Detailed Settings of Comparative Evaluation and Analysis (Section 5.1 and 5.3)

A single A100 GPU is used for experiments in Section 5.1 and 5.3.

**Setups for Unconditional Generation**    We trained the models on FFHQ [27] and CelebA-HQ [26] datasets. All training was performed with AdamW optimizer [43] with the learning rate as $1e-4$ or $2e-5$, and better results were reported. For ADM, we trained 1M iteration with batch size 8 for the FFHQ dataset and trained 400K iterations with batch size 16 for the CelebA-HQ dataset. For LDM, we trained 400K iterations with batch size 30 for both FFHQ and CelebA-HQ datasets. We generate 10K samples with a DDIM-50 step sampler and measure FID [18], Precision [36], and Recall [36] scores. For all evaluation metrics, we use all training data as reference data. FID is calculated with clean-FID [51], and Precision and Recall are computed with publicly available code [6]. All analyses are conducted above trained models.

**Setups for Class-Conditional Generation**    We trained the DiT-S/2 [52] on ImageNet dataset [7]. All training was performed with the AdamW optimizer [43] with the learning rate of $1e-4$ or $2e-5$, and better results were reported. As in DiT [52], we applied the classifier-free guidance [19] and trained 800K iterations with a batch size of 50. All samples are generated by a DDPM 250-step sampler. For evaluation metrics, we follow the evaluation protocol in ADM [8], by using their evaluation code[7]. We used the cosine schedule [47] for noise scheduling and SD-XL VAE [53] for our VAE.

## E.2  Detailed Settings of Comparison to Loss Weighting Methods (Section 5.2)

We trained the DiT-L/2 [52] on ImageNet dataset [7]. All training was performed with the AdamW optimizer [43] with the learning rate of $1e-4$. As in DiT [52], we applied the classifier-free guidance [19] and trained 400K iterations with a batch size of 256. All samples are generated by a DDPM 250-step sampler and classifier-guidance scale of $1.5$. We used the cosine schedule [47] for noise scheduling. For experiments, we used 8 A100 GPUs.

## E.3  Detailed Settings of Combining MTL Methods with Sophisticated Training Objectives (Section 5.4)

We trained three different models: vanilla LDM, vanilla LDM with P2 [6], and vanilla LDM with P2, PCgrad [83], and UW [29] applied simultaneously. All training configurations are the same in Section 5.1 but we use 500K iterations. We generate 50K samples for evaluation with a DDIM 200-step sampler and evaluate FID.

# F  Qualitative Results

In this section, we provide qualitative comparison results, which were omitted from the main paper due to space constraints. In Figure 9, 10, 11 and 12, we visualize the generated images by all models that are used for results in Table 1. As shown in the results, we can observe that incorporating MTL methods for diffusion training can improve the quality of generated images. One noteworthy observation is that UW [29] tends to generate higher-quality images compared to NashMTL [46] and PCGrad [83]. This finding aligns with the results observed in Table 1.

Moreover, we plot the randomly selected samples from 50K generated data in Fig. 13. Despite being randomly selected, the majority of the generated images exhibit remarkable fidelity.

---

[6]https://github.com/youngjung/improved-precision-and-recall-metric-pytorch
[7]https://github.com/openai/guided-diffusion/tree/main/evaluations

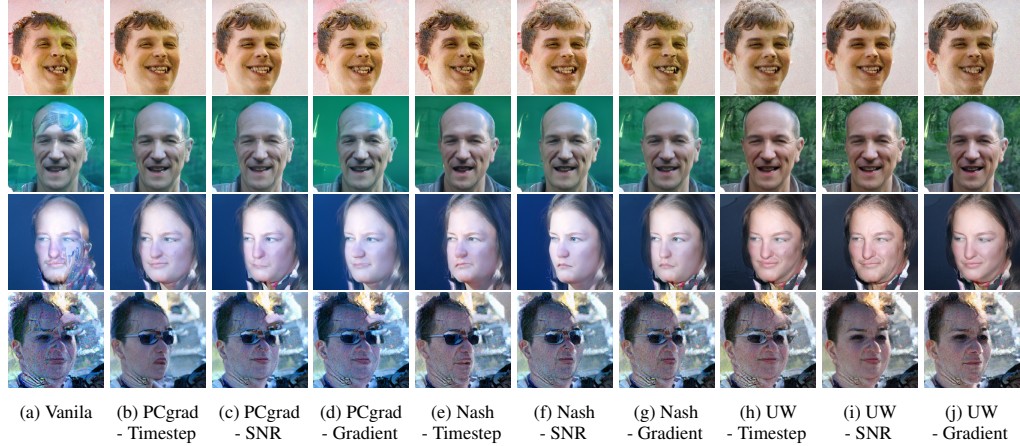

| (a) Vanila | (b) PCgrad - Timestep | (c) PCgrad - SNR | (d) PCgrad - Gradient | (e) Nash - Timestep | (f) Nash - SNR | (g) Nash - Gradient | (h) UW - Timestep | (i) UW - SNR | (j) UW - Gradient |

Figure 9: Qualitative comparison of ADM trained on the FFHQ dataset.

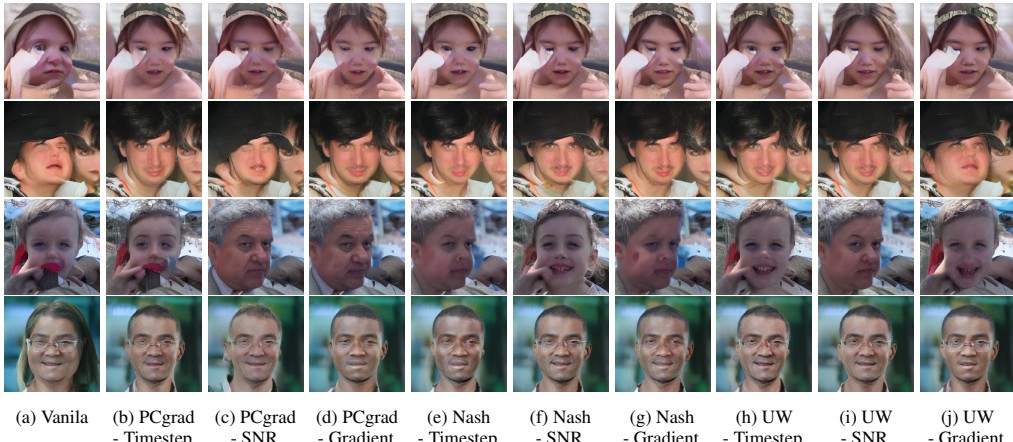

| (a) Vanila | (b) PCgrad - Timestep | (c) PCgrad - SNR | (d) PCgrad - Gradient | (e) Nash - Timestep | (f) Nash - SNR | (g) Nash - Gradient | (h) UW - Timestep | (i) UW - SNR | (j) UW - Gradient |

Figure 10: Qualitative comparison of LDM trained on the FFHQ dataset.

## G   Dynamic Programming Algorithm for Interval Clustering

In this section, we introduce the algorithm for optimizing the interval cluster and the implementation details. The optimal solution of interval clustering can be found using dynamic programming for a $L_{cluster}$ function [2, 49, 76]. The sub-problem is then defined as finding the minimum cost of clustering $\mathcal{X}_{1,i} = \{1, \ldots, i\}$ into $m$ clusters. By saving the minimum cost of clustering $\mathcal{X}_{1,i} = \{1, \ldots, i\}$ into $m$ clusters to the matrix $D[i, m]$, the value in $D[T, k]$ represents the minimum clustering costs for the original problem in Eq. 4. For some timestep $m \leq j \leq i$, $D[j - 1, m - 1]$ must contain the minimum costs for clustering $\mathcal{X}_{1,j-1}$ into $(m - 1)$ clusters [49, 76]. This establishes the optimal substructure for dynamic programming, which leads to the recurrence equation as follows:

$$D[i, m] = \min_{m \leq j \leq i} \left\{ D[j - 1, m - 1] + L_{cluster}(\mathcal{X}_{j,i}) \right\}, \quad 1 \leq i \leq T, \quad 1 \leq m \leq k. \quad (6)$$

To obtain the optimal intervals $l_1, \ldots, l_k$, we use $S[i, m]$ to record the argmin solution of Eq. 6. Then, we backtrack the solution in $O(k)$ time from $S[T, k]$ by assigning $l_m = S[l_{m+1} - 1, m]$ from $m = k$ to $m = 1$ by initializing $l_k = S[T, k]$.

Interval clustering with SNR-based or gradient-based objectives can produce unbalanced sizes of each interval, which causes unbalanced allocation of task clusters due to randomly sampled timestep $t$. Therefore, we add constraints on the size of each cluster to avoid seriously unbalanced task clusters. To add constraints on the size of each cluster $n_i = |I_i| = r_i - l_i + 1$ for $i = 1, ..., k$, we define the lower and upper bounds of it as $m_I$ and $M_I$ with $m_I \leq \frac{n_i}{k} \leq M_I$. In Eq. 6, the $m$-th cluster (i.e., $\mathcal{X}_{j,i}$) size $n_m$ must range from $m_I$ to $M_I$, yielding $i + 1 - M_I \leq j \leq i + 1 - m_I$. Furthermore, to

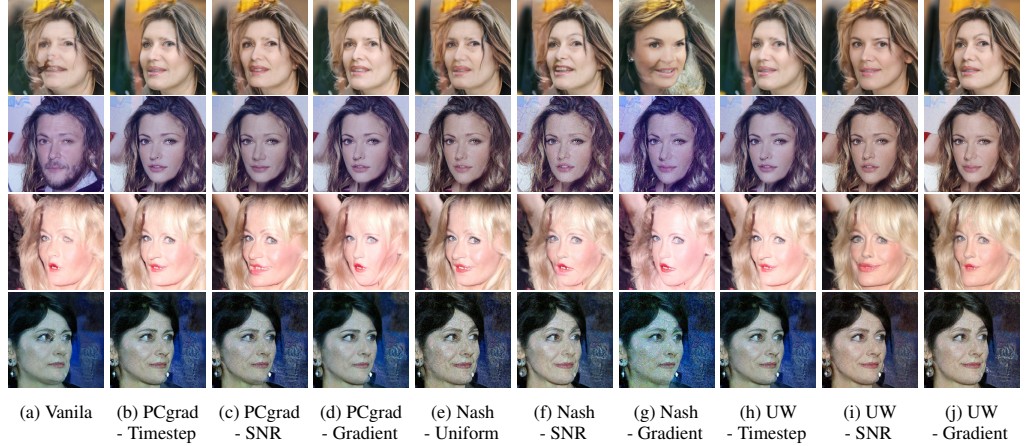

(a) Vanila  (b) PCgrad - Timestep  (c) PCgrad - SNR  (d) PCgrad - Gradient  (e) Nash - Uniform  (f) Nash - SNR  (g) Nash - Gradient  (h) UW - Timestep  (i) UW - SNR  (j) UW - Gradient

Figure 11: Qualitative comparison of ADM trained on the CelebA-HQ dataset.

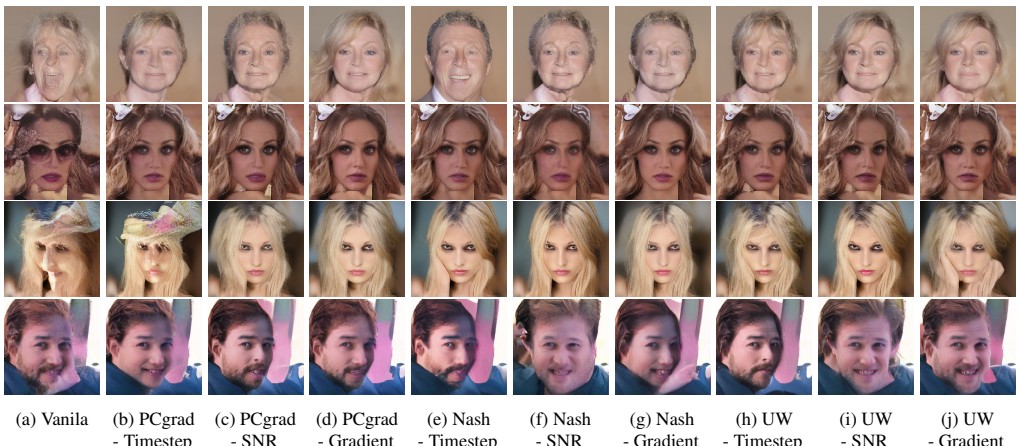

(a) Vanila  (b) PCgrad - Timestep  (c) PCgrad - SNR  (d) PCgrad - Gradient  (e) Nash - Timestep  (f) Nash - SNR  (g) Nash - Gradient  (h) UW - Timestep  (i) UW - SNR  (j) UW - Gradient

Figure 12: Qualitative comparison of LDM trained on the CelebA-HQ dataset.

satisfy the $(m-1)$-clusters constraint, $1+(m-1)m_I \leq j$. Finally, Eq. 6 with constraints on the size of the cluster is derived as follows:

$$D[i,m] = \min_{\substack{\max\{1+(m-1)m_I, i+1-M_I\} \leq j \\ j \leq i+1-m_I}} \left\{ D[j-1, m-1] + L_{cluster}(\mathcal{X}_{j,i}) \right\}, 1 \leq i \leq T, 1 \leq m \leq k. \tag{7}$$

Specifically, we assign $\lfloor \frac{T}{2k} \rfloor$ and $\lceil \frac{3T}{2k} \rceil$ to $m_I$ and $M_I$, respectively.

## H   Broader Impacts

**Revisiting Diffusion Models through Multi-Task Learning**   Our work revisits diffusion model training from a Multi-Task Learning aspect. We show that negative transfer still occurs in diffusion models and addressing it with MTL methods can improve the diffusion models. Starting from our work, a better understanding of the multi-task learning characteristics in diffusion models can lead to further advancements in diffusion models.

**Negative Societal Impacts**   Generative models, including diffusion models, have the potential to impact privacy in various ways. For instance, in the context of DeepFake applications, where generative models are used to create realistic synthetic media, the training data plays a critical role in shaping the model's behavior.

When the training data is biased or contains problematic content, the generative model can inherit these biases and potentially generate harmful or misleading outputs. This highlights the importance

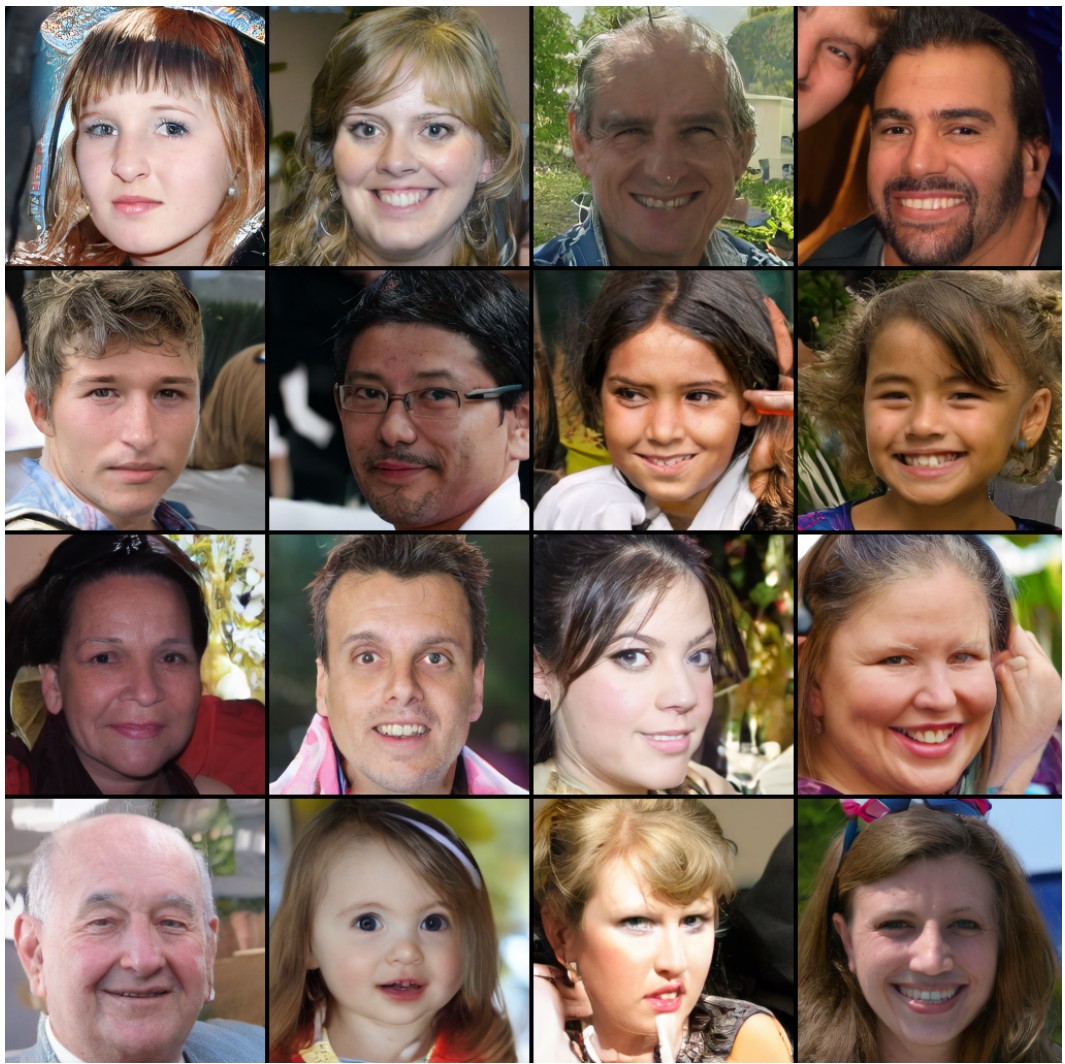

Figure 13: Randomly selected images from generated images of LDM with combined methods of UW, PCgrad, and P2 on the FFHQ dataset. DDIM 250-step sampler is used.

of carefully selecting and curating the training data for generative models, particularly when privacy and ethical considerations are at stake.

# I  Limitations

Our work has two limitations that can be regarded as future works. Firstly, we have not yet completely resolved the issue of negative transfer in the training of diffusion models as shown in Fig. 5. This indicates that learning entire denoising tasks still causes degradation in certain denoising tasks. By successfully addressing this degradation and enabling the model to harmoniously learn entire denoising tasks, we anticipate significant improvements in the performance of the diffusion model.

Secondly, our study does not delve into the architectural design aspects of multi-task learning methods. While our focus lies on model-agnostic approaches in MTL, it is worthwhile to explore the possibilities of designing appropriate architectures within an MTL framework. Previous works in diffusion models utilize timestep and noise level as input, which can be considered as using task embeddings scheme [73]. By revisiting these aspects, the architecture of the diffusion model can be further advanced in future works.

