# OpenReview forum: "Addressing Negative Transfer in Diffusion Models"
_NeurIPS.cc/2023/Conference — NeurIPS 2023 poster_

### Official Review · Reviewer_sPtA · 2023-07-06

**Soundness:** 3 good
**Presentation:** 4 excellent
**Contribution:** 3 good
**Rating:** 6
**Confidence:** 4

**Summary:**

This paper discusses the use of diffusion-based generative models for various generative tasks, such as image, video, 3D shape, and text generation. The authors argue that negative transfer, which refers to competition between conflicting tasks leading to decreased performance, should be investigated and addressed in diffusion models. They observe a negative correlation between task affinity and the difference in noise levels, suggesting that adjacent denoising tasks are more harmonious. They also find evidence of negative transfer in diffusion model training, where utilizing a model trained exclusively on denoising tasks within a specific interval generates higher-quality samples compared to a model trained on all tasks simultaneously. To address negative transfer, the authors propose leveraging existing multi-task learning techniques and clustering denoising tasks into intervals. They demonstrate the effectiveness of their approach through experiments on image datasets, showing improved quality in generated images.

**Strengths:**

1, The paper provides a comprehensive analysis of diffusion-based generative models, considering their performance and flexibility in various generative tasks, including image, video, 3D shape, and text generation. It highlights the achievements and potential areas for improvement in diffusion models.
2, The authors identify and address the issue of negative transfer in diffusion models. They observe evidence of negative transfer during model training and propose a clustering approach to mitigate its impact on denoising tasks. This analysis adds valuable insights to the field of multi-task learning and diffusion models.

**Weaknesses:**

1, The experiments conducted in the paper are limited to specific datasets (FFHQ and CelebA-HQ) and specific types of diffusion models (Ablated Diffusion Model and Latent Diffusion Model). This limited scope may restrict the generalizability of the proposed approach to other datasets or types of diffusion models. The paper could have benefited from evaluating the method on a broader range of datasets and models.

2, While the paper presents empirical results to support the effectiveness of the proposed approach, it lacks in-depth theoretical analysis or mathematical formulation of the clustering strategy. A more rigorous theoretical analysis would have provided a deeper understanding of the approach and its underlying principles.

**Questions:**

1, What is the proposed clustering approach for addressing negative transfer in diffusion models? How does it take into account noise levels and task affinity?
2, What are the limitations of the proposed clustering approach in terms of scalability and computational complexity?
3, How did the authors validate the effectiveness of their approach? Can you provide more details about the experimental setup and the results obtained?

**Limitations:**

As stated in "Weakness".

---

> ### Author Rebuttal · Authors · 2023-08-09
>
> We appreciate the insightful feedback provided by reviewer sPtA. We will address the raised concerns and revise the paper accordingly, as these comments contribute significantly to improving our work.
>
> ---
>
> ### **W1: Lack of experiments on different datasets and diffusion models.**
>
> Thank you for bringing up this point. We acknowledge the importance of validating our method across diverse models and datasets. In response, we extended our experiments to include the transformer-based DiT-S architecture on the ImageNet 256x256 dataset. The results are presented in **Fig. A-B in the Global Response**.
>
> Across different architectures and larger-scale datasets like ImageNet, our method consistently enhances FID, IS, and Precision across MTL methods compared to vanilla training. Notably, methods (UW, NashMTL) excelling in balancing are particularly effective. Additionally, convergence analysis in Fig. B highlights that our method (especially UW and NashMTL) noticeably accelerates convergence compared to vanilla training, and eventually leads to superior final performance.
>
> These experiments strongly affirm our method's generalizability. We will add these results to the final version of the manuscript.
>
> ---
>
> ### **Q1: Proposed clustering approach for addressing negative transfer.**
>
> The intuition behind task clustering is:
>
> - The clustering of tasks is necessary to apply MTL methods to diffusion models. Treating each denoising task as an independent task and applying MTL methods is simply not feasible due to instability and large computational complexity.
> - By clustering the denoising tasks into task groups, we best reap the benefits of multi-task learning. In other words, by letting the model treat similar tasks assigned to a task cluster simultaneously, we may expect that it prevents negative transfer among such tasks. Meanwhile, we use various methods suggested in MTL literature to reduce the negative transfer between tasks belonging to different task clusters.
>
> Considering these, we opt for the interval clustering method based on our observation that tasks close in timestep have higher task affinity scores. To demonstrate the effectiveness of interval clustering, we tried and applied MTL methods on top of task groups discovered by Higher Order Approximation (HOA)-based task grouping which does not constraint that tasks within a cluster must be neighboring in timesteps (See Appendix D.3). We found that interval clustering outperforms HOA, showing grouping tasks into disjoint intervals is a valid strategy.
>
> We consider timestep, SNR, and task affinity scores as clustering criteria. By considering SNR-based clustering costs, our method encourages the grouping of tasks with similar noise levels, effectively accounting for noise levels. By using task affinity-based objectives, our approach enhances task affinity among clusters, allowing for the inclusion of task affinity in cluster considerations.
>
> ---
>
> ### **W2 & Q2: In-depth theoretical analysis and computational complexity.**
>
> We appreciate your thoughtful feedback. Regarding the theoretical analysis of the clustering algorithm, we have built upon well-established principles outlined in [43, 65]. In Appendix G, we have detailed use of dynamic programming to optimize interval clusters. Let $n$ and $k$ denote the number of data points ($T$, in our case) and clusters, respectively. The interval clustering algorithm employs a matrix $D[i,m]$ to store minimum clustering costs for interval length $i$ and cluster count $m$, with a total memory cost of $O(nk)$. Each matrix element involves $O(n\omega(n))$ computation, resulting in time complexity of $O(n^2k\omega(n))$. For each of our three clustering cost functions:
>
> - **Timestep**: The calculation of cluster costs takes $O(n)$ time. Hence, this approach operates in $O(n^3k)$ time and uses $O(nk)$ memory.
> - **SNR:** Computation of SNR in constant time keeps its time complexity at $O(n^3k)$. Memory complexity is also $O(nk)$.
> - **Gradient:** Unlike the aforementioned methods, the gradient-based clustering capitalizes on precomputed task affinity scores $\mathrm{TAS}(t_x,t_y)$ for $x,y\in[1, T]$, resulting in a memory cost of $O(n^2)$. The clustering cost computation involves memory-referencing, resulting in time complexity of $O(n^3k)$ and memory $O(nk+n^2)$.
>
> Furthermore, it's worth noting that potential exists for further optimization, reducing time complexity to $O(n^2k)$, as shown in [A].
>
> ---
>
> ### **Q3: Evaluation of the Proposed Method**.
>
> For fair comparison and analysis, our experimental setup follows that of prior research such as ADM [7], P2 [6], and LDM [47]. We provide a list of all our experiments and their corresponding experimental setup details below:
>
> - Section 5.1: Incorporating MTL methods with interval clustering dramatically improves diffusion training. Setups are in Section 5 and Appendix E.1.
> - Section 5.2: 1) MTL methods exhibits fast convergence. 2) Our methods improve negative transfer in diffusion models. Setups are presented in Section 5 and Appendix E.1.
> - Section 5.3: Our method is also effective on a more sophisticated training objective, P2. Setups are presented in Section 5 and Appendix E.1.
> - Appendix D.3: Interval clustering outperforms the HOA-based task grouping method in terms of incorporating MTL methods. Setups are presented in Appendix D.3.
> - Other experimental analyses: We conducted several empirical analyses about 1) the existence of negative transfer, 2) task affinity, 3) ablation on $k$, and 4) the behavior of MTL methods.
>
> In the final version, we will clearly present the above experimental settings by making explicit references. Furthermore, we release experimental code for explicitly sharing our experimental settings.
>
> ---
>
> ### **References**
>
> [A] Grønlund, et al., “Fast exact k-means, k-medians and Bregman divergence clustering in 1D.”, arXiv 2017.

---

### Official Review · Reviewer_8Ptq · 2023-07-07

**Soundness:** 4 excellent
**Presentation:** 4 excellent
**Contribution:** 3 good
**Rating:** 7
**Confidence:** 5

**Summary:**

This paper analyzes diffusion training from an MTL perspective and observes the negative transfer in diffusion training. Several multi-task learning algorithms are employed to address the negative transfer problem, leading to improved performance.

**Strengths:**

1. The paper presents a detailed analysis of task affinity and negative transfer across different diffusion steps, which is interesting.
2. The incorporation of MTL methods has yielded a substantial improvement in the performance of the diffusion model.

**Weaknesses:**

1. The paper lacks a comparison of time and GPU memory usage between the vanilla and MTL approaches. Typically, MTL methods require more time and significantly higher GPU memory.
2. It would be beneficial to include random weights and linear scalarization baselines. Several studies have suggested that linear scalarization and random weights can serve as strong baselines [1][2].

[1] Xin et al., Do Current Multi-Task Optimization Methods in Deep Learning Even Help?, NeurIPS 2022

[2] Lin et al., A closer look at loss weighting in multi-task learning, arXiv 2022


**Questions:**

Have you considered utilizing the weights obtained by MTL approaches as the loss weights for training the model? For instance, using the weighted loss with the mean weight of each interval in Figure 3b. Would this approach yield comparable results?

**Limitations:**

See weakness.

---

> ### Author Rebuttal · Authors · 2023-08-09
>
> We deeply appreciate the insightful comments by reviewer 8Ptq. The comments are very helpful in making our work more complete from an MTL perspective. We will address all raised concerns by the reviewer and revise the paper accordingly.
>
> ---
>
> ### **W1: Lack of comparison of time and GPU memory complexity.**
>
> Thank you for your valuable suggestion. First, we point out that clustering cost is almost free compared to the cost of diffusion model training, with the exception of gradient-based clustering which requires training of a diffusion model until convergence. Next, in order to assess the additional resource requirements incurred by MTL, we measured GPU memory usage and runtime when training with the LDM architecture and timestep-based clustering on the FFHQ dataset. The results are summarized in **Table A in the General Response**.
>
> We explain the results below:
>
> - PCgrad exhibits a runtime of 1.523 iterations/second, which is comparatively slower than vanilla training's pace of 2.108 iterations/second. This discrepancy primarily arises from the inherent requirement of PCgrad to compute per-interval gradients for every iteration. However, it is worth noting that PCgrad uses less GPU memory compared to vanilla training. This stems from the partitioning of minibatch samples across intervals for per-interval gradient calculation, resulting in a reduced number of samples employed for backpropagation compared to vanilla training.
> - NashMTL exhibits a marginal decrease in runtime with 2.011 iterations/second compared to vanilla training. Similar to PCgrad, this minor difference in runtime arises due to the requirement for per-interval gradients. However, NashMTL offers a practical speed-up strategy by computing per-interval gradients every few iterations, in contrast to PCgrad. This distinction results in NashMTL outperforming PCgrad significantly in terms of runtime. However, NashMTL uses more GPU memory, which can be attributed to the caching of the parts for calculating weights.
> - UW exhibits comparable runtime and GPU memory usage to vanilla training, owing to its utilization of weighted loss and efficient weight updates through a single backpropagation pass.
>
> We also note that applying MTL methods only incurs additional compute during training, and does not affect the inference time. We will add this additional information to our final manuscript.
>
> ---
>
> ### **W2: Additional baselines.**
>
> We are truly grateful for your valuable suggestion. Indeed, Linear Scalarization (LS) and Random Loss Weighting (RLW) should serve as simple and strong baselines, and comparison of our methods to them will establish the necessity of applying such sophisticated MTL methods as PCGrad, NashMTL, and Uncertainty Weighting. Accordingly, we provide results for LS and RLW on the FFHQ dataset using ADM architecture in **Table B of the Global Response**. We explain the results below:
>
> - As seen in the results, LS achieves slightly worse performance than vanilla training, which suggests that simply re-framing the diffusion training task as an MTL task and applying a naïve MTL loss is not enough.
> - Also, RLW achieves much worse performance compared to vanilla training. It appears that the randomness introduced by the weighting interferes with diffusion training. This indicates that sophisticated weighting schemes such as UW and NashMTL indeed are responsible for significant performance gain.
>
> Overall, these additional experiments support the effectiveness of the applied MTL methodologies. We will include these results in the final manuscript.
>
> ---
>
> ### **Q1: Utilizing weights obtained by MTL approaches.**
>
> Thank you for the suggestion. We find this approach very intriguing. As suggested, we obtained the average weight assigned to each task interval (for both UW and NashMTL), upon finishing the training of LDM on FFHQ. We then re-trained the model with the fixed weights obtained above. The results are presented in **Table C in the Global Response**. For both NashMTL and UW, using fixed weights leads to a slight degradation in FID and Precision Score, which suggests that adaptive weight assignment using NashMTL and UW offers an advantage, albeit slightly in this particular case. It is also notable that despite this slight decrease, using weighted loss with fixed weights still surpasses vanilla training in terms of sample quality. We will include this result in the final manuscript.

---

> > ### Comment · Reviewer_8Ptq · 2023-08-20
> >
> > Thanks for your responses. My concerns have been addressed.

---

> > > ### Author Response · Authors · 2023-08-20
> > > **Thanks to reviewer 8Ptq**
> > >
> > > We deeply appreciate your effort in responding to the rebuttal. We are delighted that your concerns have been resolved.

---

### Official Review · Reviewer_7mPc · 2023-07-07

**Soundness:** 3 good
**Presentation:** 4 excellent
**Contribution:** 3 good
**Rating:** 5
**Confidence:** 4

**Summary:**

This work explores the phenomenon of negative transfer in the diffusion training procedure, where different time steps or signal-to-noise ratios may conflict with each other and degrade overall performance. The authors propose a solution to this problem by introducing internal clustering and implementing several multi-task learning (MTL) methods to mitigate the negative transfer effect. The proposed approach is evaluated through experiments on the FFHQ and CelebA datasets, and the results demonstrate its effectiveness in improving performance compared to existing methods.

**Strengths:**

1. It is interesting and novel to approach the training of diffusion models from the perspective of multi-task learning (MTL). By recognizing the potential for negative transfer in the diffusion training procedure, the authors of this work are able to leverage the benefits of MTL to mitigate the negative effects of conflicting time steps or signal-to-noise ratios.
2. The motivation sounds good and the preliminary experiment can demonstrate the negative transfer phenomenon in diffusion models.
3. After addressing negative transfer through internal clustering and MTL methods, the final model achieve good performance on FFHQ and CelebA.
4. The writing is good and easy to follow.


**Weaknesses:**

My main concern is that the experiments are only conducted on small datasets like FFHQ and CelebA. Now many standard benchmarks exist. e.g., ImageNet-64, ImageNet-256, CoCo. Experiments on more datasets are needed to prove the effectiveness of the proposed methods.


**Questions:**

1. It would be beneficial to include the average conflict of the baseline model in Fig 3(a) to provide a clearer comparison.
2. The choice of the number of clusters is an important parameter in the proposed internal clustering approach, and it would be beneficial to conduct further analysis and ablation studies to determine the optimal number of clusters.

---

> ### Author Rebuttal · Authors · 2023-08-09
>
> We are grateful to reviewer 7mPc for providing constructive comments, which are very helpful in improving our work through experimental results. We will address all raised concerns by the reviewer and revise the paper accordingly.
>
> ---
>
> ### **W1: Experiments are only conducted on small datasets.**
>
> Thank you for your valuable suggestion. We agree with your point that our method must be validated on a large-scale dataset. Accordingly, we conducted experiments on ImageNet 256 x 256 for the conditional generation task using the DiT-S model. We opted for a smaller architecture of DiT-S due to resource constraints.
>
> We report evaluation metrics for generated samples under varying guidance scales in **Fig. A of the Global Response**. As shown in Fig. A, our method consistently improves upon vanilla training in terms of FID, IS, and Precision scores by a large margin.  Specifically, balancing methods such as UW and NashMTL are shown to be very effective under strong guidance. We also provide convergence analysis in **Fig. B of the Global Response**. As shown in Fig. B, we see that applying our method leads to faster convergence and better final performance, across all MTL methods considered.
>
> With the additional results on ImageNet, we believe our work has gained stronger empirical results. We will include the detailed experimental setup and results in the final manuscript.
>
> ---
>
> ### **Q1: Comparison to the average conflict of the baseline in Fig. 3a.**
>
> We thank the reviewer for this valuable insight. We report the average conflict of the baseline in **Fig. C of the Global Response**. As can be seen in Fig. C, the average conflict of the baseline does not significantly differ from that of PCgrad (Fig. 3a). This result is in line with the recent findings in [B], which show that MTL methods dealing with conflicting gradients do not actually reduce the *occurrence* of conflicting gradients. We will include this additional information for clarity and completeness in the final manuscript.
>
> ---
>
> ### **Q2: Ablation studies on the number of clusters.**
>
> We apologize for not making explicit references in the main paper to the ablation experiment results in the Appendix. We would like to point the reviewer to Section D.2 of the Appendix, where we perform ablation on the number $k$ of clusters.
>
> As shown in the result, our method is robust to the number of clusters and improves upon vanilla diffusion training in all cases considered. Specifically, we see that applying MTL methods with only two clusters ($k=2$) already leads to a significant performance boost, which confirms the effectiveness of our method. For clarity and completeness, we will provide a concise summary of the ablation experiments and make an explicit reference to the results in the main paper.
>
> ---
>
> ### **References**
>
> [A] Peebles et al., “Scalable Diffusion Models with Transformers”, arXiv 2022.
>
> [B] Shi, Guangyuan, et al., “Recon: Reducing Conflicting Gradients From the Root For Multi-Task Learning.”, ICLR 2023.

---

> > ### Comment · Reviewer_7mPc · 2023-08-18
> >
> > thank authors for the rebuttal. My concerns are all addressed and I keep my rating.

---

> > > ### Author Response · Authors · 2023-08-18
> > > **Thanks to reviewer 7mPc**
> > >
> > > Thank you for taking the time to respond to the rebuttal.
> > > We are glad that your concerns have been resolved.

---

### Author Rebuttal · Authors · 2023-08-09

# Global Response

Dear reviewers,

We sincerely thank you for dedicating time and effort to review our manuscript. In an attached PDF file, we have provided the results of all conducted experiments during the author response period for addressing concerns and questions raised by reviewers. In this response, we offer a concise explanation and setup of experiments in the attached PDF file.

---

## Contents

- **[Figure A and B]**: To address concerns raised by reviewers 7mPc & sPtA, we conducted experiments on the large-scale dataset, ImageNet 256x256, with transformer-based diffusion models, DiT-S [A]. All training was performed for 800K iterations with AdamW optimizer with a learning rate of 1e-4 or 2e-5, and better results were reported. Figure A illustrates FID, IS, Precision, and Recall scores according to the classifier-free guidance scale from 1.5 to 3.0. Figure B depicts the trajectory of the FID score over training iterations with the classifier-free guidance scale 2.0. All samples are generated by DDPM 250-step sampler.
- **[Figure C]**: To answer question 2 of reviewer 7mPc, we plot the average number of gradient conflicts in baseline training (referred to as ‘vanilla’ in the manuscript). The settings are the same as in Fig. 3a of the manuscript.
- **[Table A]**: To address concerns raised by reviewer 8Ptq, a comparison of GPU memory usage and runtime for all methods is presented in Table A. These measurements were conducted using the setting where the number of clusters $k$ was set to 5 on the FFHQ dataset with LDM and timestep-based clustering.
- **[Table B]**: For reflecting reviewer 8Ptq’s suggestion, the results of Linear Scalarization (LS) [B] and Random Loss Weighting (RLW) [C] on the FFHQ dataset with ADM architecture are provided in Table B. All settings are the same as in Table 1 of the manuscript.
- **[Table C]:** To respond to the question of reviewer 8Ptq, we applied the weighted loss with the mean weight of each interval of UW and NashMTL in Fig. 3b and Fig. 3c and report the result in Table C. All experimental settings are the same in Figure 3 of the manuscript.

---

### References

[A] Peebles et al., “Scalable Diffusion Models with Transformers”, arXiv 2022.

[B] Xin et al., “Do Current Multi-Task Optimization Methods in Deep Learning Even Help?”, NeurIPS 2022

[C] Lin et al., “A closer look at loss weighting in multi-task learning”, arXiv 2022

---

### Decision · Program_Chairs · 2023-09-21

**Decision:**

Accept (poster)

**Comment:**

All reviewers liked the paper and engaged well in the reviewing and discussion phase. Accept is recommended.